# The Impact of Patellar Tendon Advancement on Knee Joint Moment and Muscle Forces in Patients with Cerebral Palsy

**DOI:** 10.3390/life11090944

**Published:** 2021-09-09

**Authors:** Derya Karabulut, Yunus Ziya Arslan, Marco Götze, Sebastian I. Wolf

**Affiliations:** 1Department of Mechanical Engineering, Faculty of Engineering, Istanbul University-Cerrahpaşa, Avcilar, Istanbul 34320, Turkey; derya.karabulut@iuc.edu.tr; 2Department of Robotics and Intelligent Systems, Institute of Graduate Studies in Science and Engineering, Turkish-German University, Beykoz, Istanbul 34820, Turkey; yunus.arslan@tau.edu.tr; 3Centre of Orthopedics and Trauma Surgery, Universitätsklinikum Heidelberg, 69118 Heidelberg, Germany; Marco.Goetze@med.uni-heidelberg.de

**Keywords:** cerebral palsy, crouch gait, patellar tendon advancement, knee muscle forces, knee joint moment, modeling and simulation

## Abstract

Background: Patellar tendon advancement (PTA) is performed for the treatment of crouch gait in patients with cerebral palsy (CP). In this study, we aimed to determine the influence of PTA in the context of single-event multilevel surgery (SEMLS) on knee joint moment and muscle forces through musculoskeletal modeling; Methods: Gait data of children with CP and crouch gait were retrospectively analyzed. Patients were included if they had a SEMLS with a PTA (PTA group, *n* = 18) and a SEMLS without a PTA (NoPTA group, *n* = 18). A musculoskeletal model was used to calculate the pre- and postoperative knee joint moments and muscle forces; Results: Knee extensor moment increased in the PTA group postoperatively (*p* = 0.016), but there was no statistically significant change in the NoPTA group (*p* > 0.05). The quadriceps muscle forces increased for the PTA group (*p* = 0.034), while there was no difference in the NoPTA group (*p* > 0.05). The hamstring muscle forces increased in the PTA group (*p* = 0.039), while there was no difference in the NoPTA group (*p* > 0.05); Conclusions: PTA was found to be an effective surgery for the treatment of crouch gait. It contributes to improving knee extensor moment, decreasing knee flexor moment, and enhancing the quadriceps and hamstring muscle forces postoperatively.

## 1. Introduction

Patella alta, a superiorly displaced patella, is often observed and nearly universal in patients with cerebral palsy (CP) and crouch gait [1,2,3]. Crouch gait, a common pathological gait pattern in patients with CP, is characterized by abnormal kinetics, kinematics, and muscle function [3,4,5,6]. Single-event multilevel surgery (SEMLS) is the commonly preferred strategy for the surgical treatment of CP patients. Patellar tendon advancement (PTA) is generally added to the surgery design of the patients with CP and crouch gait because of their knee flexion deficits [1,5,7].

PTA, performed by patellar tendon shortening or distal advancement of the tibial tuberosity, corrects quadriceps insufficiency to enhance knee extension moment during gait by improving the quadriceps moment arm. Several studies investigated the postoperative results of the PTA using gait analysis [1,8,9,10,11]. All these studies have shown that joint kinematics, especially knee joint kinematics, improved postoperatively (please see Figure A1). Brandon et al. reported that knee flexion moment significantly increased with the proportion of severity of crouch gait [12]. Boyer et al. showed that patients subjected to distal femoral extension osteotomy (DFEO) + PTA had higher peak knee extensor moment than those who were not managed with DFEO + PTA [8]. Hyer et al. reported that DFEO with patellar tendon imbrication significantly decreased knee moment in late stance [13].

In addition to knowledge on the effect of PTA on knee joint kinematics and kinetics, determination of the knee muscle forces would also deepen our understanding of the efficacy of surgical treatment. Clinicians need to identify muscles that can be surgically lengthened, strengthened, or otherwise treated to provide a more efficient walking pattern in CP patients. Even though the function of the corresponding muscles during crouch gait is patient-specific, certain muscles are generally the target of operations for the treatment of crouch gait [14,15,16]. Das et al. [17] (by using the manual muscle testing method) and Stout et al. [1] (with the help of clinical examination) showed that the quadriceps strength was improved when PTA was performed.

Since the quantification of in vivo muscle forces requires invasive techniques, which are ethically questionable, computational musculoskeletal modeling and simulation based on experimental gait data provide a practical approach for calculating muscle forces [18,19,20]. Musculoskeletal models are integrated with quantitative motion analysis data to generate the simulation of the corresponding motion. Muscle forces can be calculated using optimization-based techniques, such as static optimization, in which a specific cost function is optimized independently at each time of movement [21].

The effect of PTA on knee joint kinematics was shown in previous studies [1,9,11,17]. However, knowledge about the postoperative effect of PTA on muscle forces is quite limited. Lenhart et al. revealed that patella position had a significant influence on the torque producing capacity of the quadriceps by using musculoskeletal modeling and simulation software [22].

In this study, to deepen our understanding of the effect of PTA on crouch walking, we aimed to quantitatively observe the effect of PTA in combination with SEMLS on the knee joint moment and muscle forces in children with CP and crouch gait through musculoskeletal modeling and simulation.

## 2. Materials and Methods

### 2.1. Study Design

The database of the local University Hospital between 2002 and 2019 was considered in this retrospective study. This study was approved by the local ethical committee of the Faculty of Medicine (S-515/2019).

Two groups were formed. Children with CP were included if they had a SEMLS in combination with a PTA (PTA group) and a SEMLS without a PTA (NoPTA group). Two procedures of the PTA surgery, namely soft tissue and bony procedures, were considered. The bony procedure of PTA was performed by the distal advancement of the tibial tuberosity. The soft tissue procedure of the PTA was executed by patellar tendon shortening. Inclusion criteria for both groups were (i) ability to walk without assistive devices or physical assistance of another person and (ii) having gait, electromyography (EMG), and X-ray data for the preoperative and postoperative periods.

Among 458 patients with CP, 21 children were nominated for the PTA group and 34 children for the NoPTA group (Figure 1). In order to ascertain the isolated effect of PTA in the context of SEMLS, patients for the PTA and NoPTA groups were determined such that they were as similar as possible in terms of (i) age, (ii) maximum knee extension at stance, (iii) maximum knee flexion at swing, (iv) knee range of motion, (v) GMFCS level, and (vi) gait profile score [23] preoperatively. Major surgeries, namely femoral extension and derotation osteotomies, hamstring lengthening, and rectus femoris transfer, applied to the patients that would affect the postoperative consequences primarily for both groups were also approved to be the same for both groups, except for the PTA surgery. The preoperative surgeries applied to the patients are provided in Table 1. In the patient selection process, 18 patients (29 limbs) for the PTA group and 18 patients (26 limbs) for the NoPTA group were considered for further analysis. A flow chart for the determination of the PTA and NoPTA groups is given in Figure 1.

### 2.2. Experimental Protocol

Temporospatial, EMG, kinematics, and kinetics gait data of the patients at self-selected walking speed were collected for the pre- and postoperative situations. The postoperative measurements were carried out 17.5 ± 5.4 months after surgery. A total of 19 reflective passive markers were attached to specific anatomical landmarks of the patients according to a standard protocol (Plugin Gait; Oxford Metrics, Oxford, UK). Three-dimensional marker data were recorded with a 12-camera Vicon motion analysis system (Oxford Metrics, Oxford, UK) at a 120 Hz sampling frequency. The ground reaction force was collected using three force plates (Kistler Instruments, Winterthur, Switzerland) with a 120 Hz sampling frequency. Surface EMG of the biceps femoris (BF), lateral gastrocnemius (LG), semimembranosus (SM), rectus femoris (RF), and vastus lateralis (VL) was recorded using a Myon System (Myon AG, Schwarzenberg, Switzerland). Linear envelopes of the full-wave rectified EMG signals treated with a 4th order Butterworth low-pass filter operating at a cut-off frequency of 6 Hz were calculated. X-ray measurements from lateral and medial projections were performed before and after surgery. According to clinical standards, the lateral view was taken at a knee flexion angle of around 30°. Moreover, gait analyses of 18 healthy children (age 13.9 ± 1.3) were performed to obtain normative data.

### 2.3. Musculoskeletal Modeling and Simulation

Rajagopal’s full-body musculoskeletal model available in the OpenSim model library was used [24]. The model with 39-degrees-of-freedom consisted of 22 segments and 80 Hill-type muscle-tendon units. First, the generic musculoskeletal model was scaled to the anthropometry of each patient such that the distances between the locations of the virtual and experimental markers were minimized. Joint angles were calculated from the marker data recorded during walking using the inverse kinematics approach [19]. Then inverse-dynamics approach was executed to compute the knee joint moments [25]. Static optimization was employed to calculate the individual muscle forces during the gait cycle. The cost function was the minimization of the sum of the squares of all muscle activations subject to the force-length and force-velocity properties of the muscles [26]. The BF muscle force was calculated as the mean value of the forces of the BF short head and long head.

### 2.4. Data Analysis

To quantitatively evaluate the difference between the joint moments and muscle forces of the PTA, NoPTA, and age-matched healthy groups, root-mean-square differences (RMSDs) and Pearson cross-correlation coefficients (PCCs) were calculated. An RMSD value of 1 indicates a mean error of 100% between the groups, while a PCC value of 1 indicates the correlation of 100% between the groups [20].

Statistical analysis was carried out by using SPSS software (Version 21.0; SPSS; Chicago, IL, USA). The statistical significance level was considered 0.05. Normality of all variables (RMSD and PCC values of the knee joint moment, ankle joint moment, and the muscle forces calculated between the PTA and healthy groups and between the NoPTA and healthy groups) was evaluated by using the Shapiro–Wilk test. They were not normally distributed. Two different statistical analyses were performed: intra-comparison for the dependent parameters (comparison of each group’s preoperative and postoperative parameters) and inter-comparison for the independent parameters (comparison of the groups with each other). Preoperative and postoperative outputs of each group were statistically analyzed by using the one-way repeated-measures ANOVA. Mann–Whitney U test was employed for the comparison of the PTA and NoPTA groups. More details regarding the statistical analyses are given in Table A1.

## 3. Results

The knee joint moments of the PTA and NoPTA groups for the pre- and postoperative exams were given in Figure 2. Magnitudes of the knee joint moments were normalized to the mass of each patient. The PTA and NoPTA groups showed similar moment patterns preoperatively (*p* > 0.05) (Figure 2a). The postoperative knee joint moment of the PTA group was closer to the healthy individuals than that of the NoPTA group (*p* = 0.018) (Figure 2b).

The ankle joint moments of the PTA and NoPTA groups for the pre- and postoperative periods were also compared (Figure 3). For the preoperative term, the moment patterns of the PTA and NoPTA groups were significantly different (*p* = 0.041) (Figure 3a). The PTA group was closer to the healthy group than that of the NoPTA group in terms of ankle joint moment postoperatively (*p* = 0.046) (Figure 3b).

In terms of the mean RMSD and PCC values of knee joint moment, the PTA group was closer to the age-matched healthy individuals than the NoPTA group postoperatively (Table 2). The mean RMSD value, indicating the level of difference between CP patients and healthy subjects, of the PTA group was 1.65 preoperatively, while it was 1.71 for the NoPTA group (*p* > 0.05). The postoperative RMSD values were 0.55 and 1.49 for the PTA and NoPTA groups, respectively (*p* = 0.018). For the PTA group, RMSD values significantly diminished (*p* = 0.016) and PCC significantly improved (*p* = 0.029) postoperatively. The RMSD and PCC values did not differ for the NoPTA group (*p* > 0.05) postoperatively.

For the ankle joint moment, the PTA group was closer to the healthy individuals than the NoPTA group postoperatively in terms of the RMSD and PCC values (Table 2). The RMSD value between the healthy and PTA groups significantly decreased postoperatively (*p* = 0.036), but the PCC value did not differ (*p* > 0.05). Between the healthy and NoPTA groups, the RMSD value did not change (*p* > 0.05) postoperatively, while the PPC value significantly increased (*p* = 0.043).

To evaluate the accuracy of the model-predicted muscle forces of the patient groups, we compared the model-predicted muscle activations and experimentally obtained EMG data (Figure 4 and Figure 5). The magnitudes of EMG data were normalized to each corresponding peak value. It can be revealed from Figure 4 and Figure 5 that the model-predicted muscle activation patterns were generally consistent with experimental EMG signals.

PCC values between the calculated muscle activations and their experimental EMG equivalents were given in Table 3. The least PCC value (0.32) was obtained from the VL muscle of the NoPTA group preoperatively, while the highest PCC value (0.95) was acquired from the RF muscle of the PTA group postoperatively.

Muscle forces calculated for the PTA, NoPTA, and healthy groups were given in Figure 6. Magnitudes of the muscle forces were normalized to the body weight (BW) of each patient. Experimental EMG data of the muscles were provided to validate the activation time-histories of muscle forces. BF muscle forces of the PTA and NoPTA groups were higher compared to the healthy group preoperatively, except for the late stance. For the postoperative period, muscle force magnitudes in the patient groups diminished. LG forces of the PTA and NoPTA groups were lower than that of healthy subjects preoperatively, while both groups’ muscle forces increased in the postoperative period. RF forces of the PTA and NoPTA groups were lower compared to the healthy group preoperatively. For the postoperative period, both groups’ muscle forces increased. SM forces of the PTA and NoPTA groups were higher than that of healthy individuals preoperatively, except for the end of the swing, while patient groups’ muscle forces decreased postoperatively. VL forces of the PTA and NoPTA groups were lower than that of the healthy group preoperatively. For the postoperative term, the VL muscle force of the PTA group was higher than that of the healthy group, while the VL muscle force of the NoPTA group was lower compared to the healthy subjects. The predicted muscle forces were generally consistent with EMG activation periods.

The RMSD and PCC values between the muscle forces of CP patients and healthy individuals were given in Table 4 and Table 5, respectively. RMSD values were higher than 0.25 for all five muscles pre- and postoperatively. PCC values were less than 0.90 for all five muscles pre- and postoperatively. The least RMSD value (0.29) was obtained from the SM muscle of the PTA group postoperatively, while the highest RMSD value (0.81) was acquired from the BF muscle of the NoPTA group preoperatively. The least PCC value (0.44) was obtained from the BF muscle of the NoPTA group preoperatively, while the highest PCC value (0.89) was acquired from the SM muscle of the PTA group postoperatively.

## 4. Discussion

PTA is a frequently preferred surgery with the combination of SEMLS for the treatment of crouch gait by improving knee extensor moment arm, and hence, by improving knee extensor moment. An objective investigation of the effects of PTA surgery on the muscle force and knee extensor moment is critical to improving the surgical strategy.

In our previous study [11], we reported that PTA effectively enhanced the Insall-Salvati ratio [27] (Table A2) and knee joint kinematics (Figure A1) of the patients with CP and crouch gait. The ankle plantar flexor-knee extensor couple is critical in understanding the relationship between the knee and ankle joints. The ankle position has a substantial role in the assignment of the knee position. In the case of weak plantar flexor muscles, the ankle generates a plantar flexor moment that maintains the ground reaction force vector anterior to the knee joint. In this way, the knee joint changes its position to be more flexed and prevents itself from collapsing. Therefore, a weak plantar flexor mechanism contributes to crouch gait [28,29]. In addition to the knee kinematics and kinetics, it would be valuable to evaluate the joint angles (Figure A2) and joint moments (Figure 3) at the ankle joint in the context of SEMLS since the ankle dorsi/plantar flexor-knee flexor/extensor couple is the main determinant for the effectiveness of the surgeries for the correction of crouch gait.

Preoperative knee joint moments of the PTA and NoPTA groups significantly differed from healthy subjects. The magnitudes of the moments for both patient groups were significantly lower than reference at the stance phase (first knee extensor moment), while the moment values of both groups were higher at the end of stance (maximum knee flexor moment) (Figure 2 and Table 2). This preoperative condition confirmed previous research in which patients with crouch gait showed insufficient knee extensor moment during walking [1,17,22]. Therefore, Sutherland and Cooper offered to diminish the knee flexion contracture for the effective treatment of crouch gait [30]. We found that the RMSD value of the knee moment for the patients with PTA significantly decreased postoperatively, while the NoPTA group still had an RMSD value of 1.49 when compared to the healthy group. This finding demonstrated that the PTA group was close to the walking pattern of healthy individuals in the postoperative period, while the NoPTA group continued to have a crouch gait pattern. Brandon et al. indicated that knee flexion moment decreased as the severity of crouch gait diminished [12]. Boyer et al. reported that peak knee extensor moment significantly increased in patients having SEMLS with PTA than those who were not subjected to PTA [8]. However, it should be noted that the PTA group proceeded to show slightly higher knee moment than healthy subjects during the end of the stance phase (maximum knee flexor moment). This finding indicated that the PTA group may have continued to have a mild flexed knee pattern.

Ankle joint moments in the PTA and NoPTA groups significantly differed from healthy subjects preoperatively (Figure 3 and Table 2). The PTA group showed inadequate dorsiflexor moment during the stance phase preoperatively, but it increased postoperatively. The NoPTA group showed excessive ankle dorsiflexor moment during the first half of the stance phase at preoperative and postoperative measurements. This finding indicated that the NoPTA group continued to have excessive dorsiflexion that may be considered as the possible cause of the crouch gait.

Experimental EMG activities were used to evaluate the accuracy of the model-predicted muscle activations and muscle forces (Figure 4, Figure 5 and Figure 6). It can be revealed that time-histories of the model-predicted muscle activations and forces of all five muscles agreed well with experimental EMG data.

The weakness of the quadriceps muscles is considered the primary reason for the crouch gait [15,31]. Sutherland and Cooper offered to restore the quadriceps muscle strength to overcome crouch gait [30]. Das et al. [17] and Stout et al. [1] reported that the quadriceps strength was advancing when PTA was performed. We found that the muscle forces of the RF and VL (quadriceps muscles) for the PTA and NoPTA groups were lower than that of the healthy group preoperatively (Figure 6). For the postoperative period, both groups improved quadriceps muscle forces. The postoperative RF muscle force of the PTA group was closer to the healthy group; the RMSD value of the RF muscle significantly decreased for the PTA group, but it did not change for the NoPTA group. This finding was also confirmed by the knee joint moment results since the knee extensor moment significantly increased in the PTA group during the postoperative period. Both patient groups provided muscle force patterns that were closer to the healthy group postoperatively (Table 5). Another result was that the VL muscle force did not significantly change for the PTA and NoPTA groups. However, for the PTA group, it was lower than healthy subjects preoperatively, while it was higher than that of healthy subjects postoperatively.

The overactivity of the hamstring muscles is seen as another possible reason for the crouch gait pattern [32]. The BF and SM (hamstring muscles) muscle forces were higher than that of the healthy group at the beginning of stance and late swing in the patient groups preoperatively (Figure 6). For the postoperative term, the hamstring muscle forces significantly decreased in the PTA group, but it did not change for the NoPTA group (Table 4). However, the PTA and NoPTA groups maintained higher BF muscle force at the beginning of the stance. In contrast to the NoPTA group, the PTA group showed higher SM muscle force than healthy subjects postoperatively. It was consistent with the finding that the PTA group still showed slightly higher knee flexor moments during gait.

The LG is the biarticular muscle that spans both knee and ankle joints, which is responsible for knee flexion and plantar flexion. Gage reported that the weakness of the plantar flexor muscles is one of the reasons for the crouch gait pattern [33]. The LG muscle forces of the PTA and NoPTA groups were lower than that of the healthy group preoperatively (Figure 6, Table 4). However, the characteristics of the muscle force curves of both groups were close to that of healthy subjects (Table 5). For the postoperative period, the LG muscle force of the PTA group significantly increased, while there was no change in the NoPTA group.

Limitations of this study should be stated. First, the generic full-body musculoskeletal model [24] used in our study was not representing the skeletal deformities of the patients. Patient-specific models, including musculoskeletal deformities, would improve the accuracy of the results. Especially the modeling of PTA surgery by representing the preoperative and postoperative patella positions on the musculoskeletal model would play a critical role in the calculation of the muscle forces. Second, generic model parameters for the muscles were used for all patients since we did not have a data set that would represent the actual architectural, contractile, and functional properties of the patients’ muscles. Third, since the patients with CP show a wide heterogeneity in terms of functional capabilities, musculoskeletal deformities, and treatment histories, we applied a rigorous matching process for inclusion and exclusion of the patients, which led to a limited number of patients in the study. For example, the patients using assistive devices during pre- or postoperative exams were excluded since the use of these devices interfere with the ground reaction force, and hence, joint moments. Under these circumstances, model-predicted muscle forces may not reflect the actual situation for the patients with assistive devices. However, the effect of PTA in combination with SEMLS on the muscle forces of CP patients using assistive devices should also be investigated through subject-specific musculoskeletal models. Lastly, we provided the short-term results of the patients. Boyer et al. reported that significant changes may develop at long-term follow-up [6]. Therefore, the results of our study should be cautiously interpreted for the long-term projections.

## 5. Conclusions

PTA was found to be an effective surgery for the restoration of crouch gait. It contributes to improving knee moment at the beginning of the stance phase (knee extensor moment), decreasing knee moment at the late stance phase (knee flexor moment), and enhancing the quadriceps and hamstring muscle forces postoperatively.

## Figures and Tables

**Figure 1 life-11-00944-f001:**
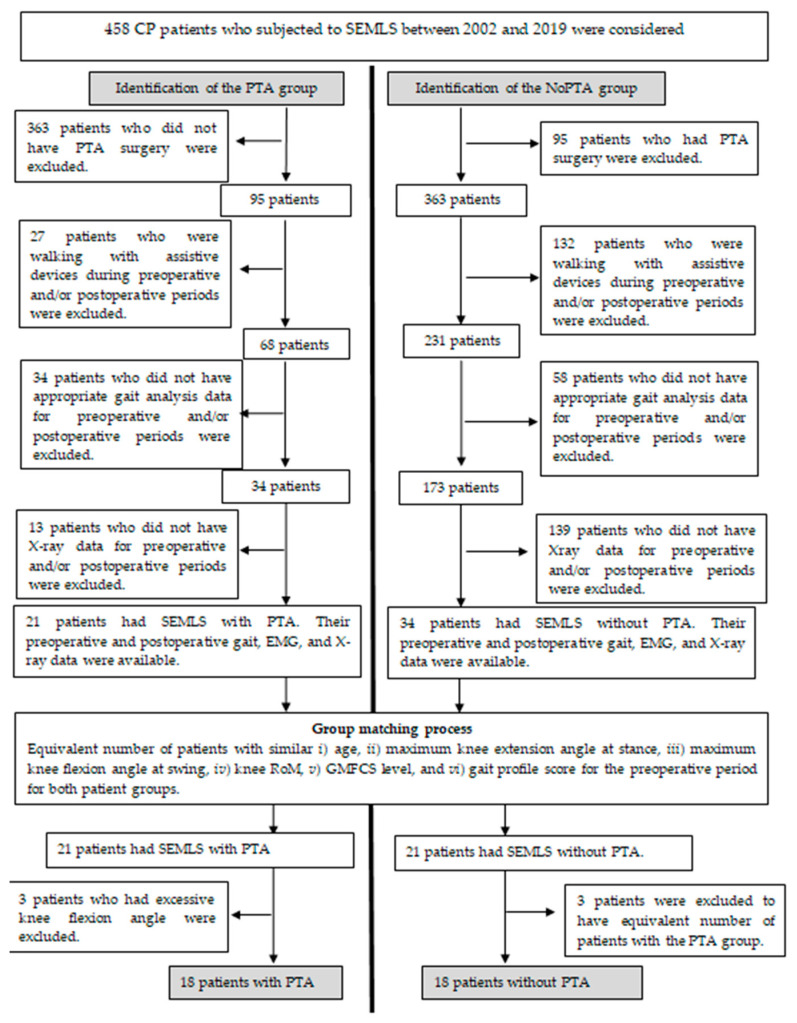
Flow chart of the patient groups. The PTA group: CP patients had single-event multilevel surgery (SEMLS) with patellar tendon advancement (PTA). The NoPTA group: CP patients had SEMLS without PTA.

**Figure 2 life-11-00944-f002:**
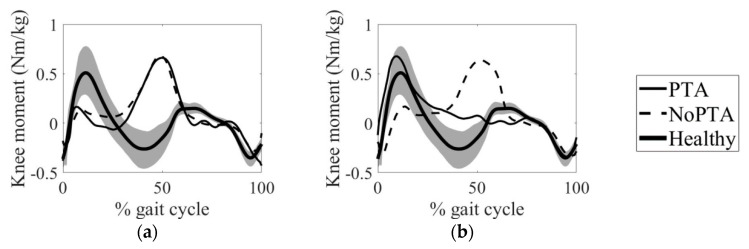
Mean knee joint flexor/extensor moment over a gait cycle for (**a**) preoperative and (**b**) postoperative periods. Gray zone indicates normative data obtained from the age-matched healthy reference group. PTA: Patellar tendon advancement, NoPTA: without patellar tendon advancement.

**Figure 3 life-11-00944-f003:**
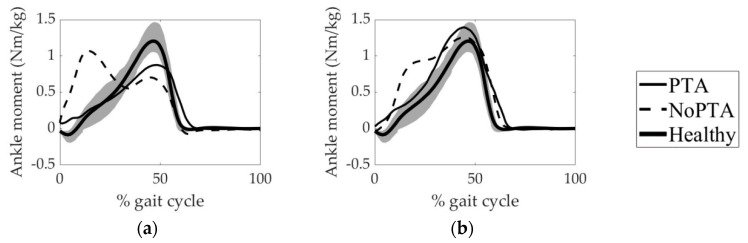
Mean plantar/dorsiflexor moment at the ankle joint over a gait cycle for (**a**) preoperative and (**b**) postoperative periods. Gray zone indicates normative data obtained from the age-matched healthy reference group. PTA: Patellar tendon advancement, NoPTA: without patellar tendon advancement.

**Figure 4 life-11-00944-f004:**
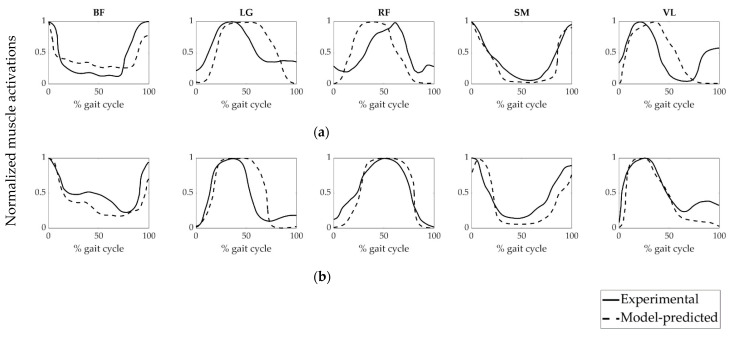
Muscle activation patterns of the PTA group. (**a**) preoperative and (**b**) postoperative muscle activations. BF: biceps femoris, LG: lateral gastrocnemius, RF: rectus femoris, SM: semimembranosus, VL: vastus lateralis. PTA: patellar tendon advancement, Experimental: linear envelope of the experimental EMG signal of the PTA group normalized to each corresponding peak value, Model-predicted: muscle activation of the PTA group predicted by static optimization.

**Figure 5 life-11-00944-f005:**
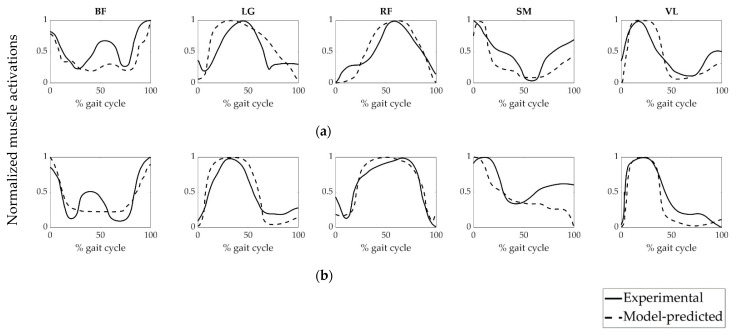
Muscle activation patterns of the NoPTA group. (**a**) preoperative and (**b**) postoperative muscle activations. BF: biceps femoris, LG: lateral gastrocnemius, RF: rectus femoris, SM: semimembranosus, VL: vastus lateralis. NoPTA: without patellar tendon advancement, Experimental: linear envelope of the experimental EMG signal of the NoPTA group normalized to each corresponding peak value, Model-predicted: muscle activation of the NoPTA group predicted by static optimization.

**Figure 6 life-11-00944-f006:**
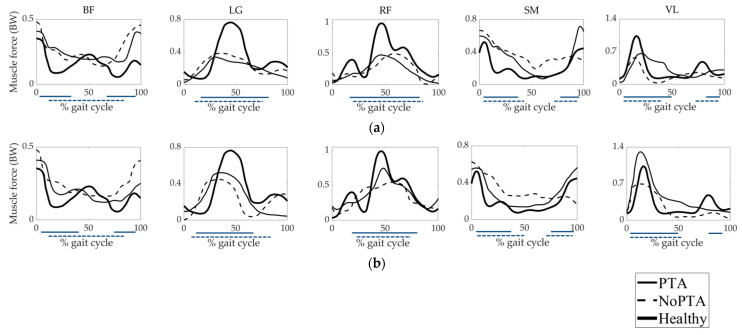
Muscle forces of the patients with PTA and without PTA, and healthy individuals. (**a**) preoperative and (**b**) postoperative muscle forces. The horizontal bars indicate the periods of experimental EMG activities. Horizontal solid blue line: PTA group, horizontal dashed blue line: NoPTA group. PTA: patellar tendon advancement, NoPTA: without patellar tendon advancement, BF: biceps femoris, LG: lateral gastrocnemius, RF: rectus femoris, SM: semimembranosus, VL: vastus lateralis.

**Table 1 life-11-00944-t001:** Preoperative surgeries for the PTA and NoPTA groups.

Surgical Procedures	PTA Group (SEMLS With PTA)	NoPTA Group (SEMLS Without PTA)
Extension osteotomy	12 patients/21 limbs	10 patients/14 limbs
Hamstring lengthening	7 patients/12 limbs	6 patients/9 limbs
Rectus femoris transfer	4 patients/8 limbs	5 patients/10 limbs
Derotation osteotomy	18 patients/30 limbs	18 patients/29 limbs

PTA: patellar tendon advancement, NoPTA: without patellar tendon advancement, SEMLS: single-event multilevel surgery.

**Table 2 life-11-00944-t002:** Mean RMSD and PCC values calculated between knee joint moments and ankle joint moments of CP patients and healthy subjects. Significant differences are marked in bold.

		Knee Joint Moment		Ankle Joint Moment	
		RMSD	PCC	RMSD	PCC
Preop	PTA vs. healthy	1.65	0.13	0.41	0.79
	NoPTA vs. healthy	1.71	0.09	0.66	0.61
Postop	PTA vs. healthy	0.55	0.77	0.31	0.88
	NoPTA vs. healthy	1.49	0.14	0.59	0.69
		*p*-value			
PTA vs. NoPTA	Preop	0.241	0.158	**0.046**	**0.041**
	Postop	**0.018**	**0.029**	**0.046**	**0.039**
preop vs. postop	PTA group	**0.016**	**0.029**	**0.036**	0.091
	NoPTA group	0.055	0.158	0.056	**0.043**

PTA: Patellar tendon advancement, NoPTA: without patellar tendon advancement, Preop: preoperative, Postop: postoperative. RMSD: Root-mean-square difference, PCC: Pearson cross-correlation coefficient.

**Table 3 life-11-00944-t003:** Mean PCC values between model-predicted muscle activations and experimental EMG data.

		BF	LG	RF	SM	VL
Preoperative	PTA	0.89	0.81	0.75	0.92	0.34
	NoPTA	0.76	0.84	0.85	0.77	0.32
Postoperative	PTA	0.93	0.92	0.95	0.93	0.88
	NoPTA	0.77	0.94	0.91	0.73	0.86

PTA: patellar tendon advancement, NoPTA: without patellar tendon advancement, BF: biceps femoris, LG: lateral gastrocnemius, RF: rectus femoris, SM: semimembranosus, VL: vastus lateralis.

**Table 4 life-11-00944-t004:** Mean RMSD values representing the difference in magnitude between muscle forces calculated for CP patients and healthy subjects. Significant differences are marked in bold.

		BF	LG	RF	SM	VL
Preop	PTA	0.69	0.52	0.58	0.48	0.38
	NoPTA	0.81	0.43	0.49	0.56	0.43
Postop	PTA	0.47	0.37	0.34	0.29	0.49
	NoPTA	0.70	0.45	0.41	0.58	0.34
		*p*-value				
PTA vs. NoPTA group	Preop	0.089	0.286	0.291	0.275	0.301
	Postop	**0.031**	0.249	0.218	**0.029**	**0.042**
Preop vs. postop	PTA group	**0.039**	**0.042**	**0.034**	**0.041**	0.063
	NoPTA group	0.065	0.318	0.305	0.315	0.062

PTA: patellar tendon advancement, NoPTA: without patellar tendon advancement, Preop: preoperative, Postop: postoperative. BF: biceps femoris, LG: lateral gastrocnemius, RF: rectus femoris, SM: semimembranosus, VL: vastus lateralis.

**Table 5 life-11-00944-t005:** Mean PCC values between muscle forces calculated for CP patients and healthy subjects. Significant differences are marked in bold.

		BF	LG	RF	SM	VL
Preop	PTA	0.45	0.85	0.61	0.85	0.75
	NoPTA	0.44	0.88	0.85	0.81	0.88
Postop	PTA	0.76	0.84	0.80	0.89	0.79
	NoPTA	0.56	0.77	0.82	0.83	0.85
		*p*-value				
PTA vs. NoPTA group	Preop	0.314	0.314	**0.029**	0.308	**0.047**
	Postop	**0.038**	0.059	0.315	0.062	**0.049**
Preop vs. postop	PTA group	**0.027**	0.308	**0.039**	0.059	0.328
	NoPTA group	0.059	**0.044**	0.064	0.064	0.063

PTA: patellar tendon advancement, NoPTA: without patellar tendon advancement, Preop: preoperative, Postop: postoperative. BF: biceps femoris, LG: lateral gastrocnemius, RF: rectus femoris, SM: semimembranosus, VL: vastus lateralis.

## Data Availability

Not applicable.

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
