# Peer review of "The Impact of Patellar Tendon Advancement on Knee Joint Moment and Muscle Forces in Patients with Cerebral Palsy"

_life, 2021, doi:10.3390/life11090944_

Round 1

Reviewer 1 Report

I would like to commend the authors on a very well designed, executed and interpreted study. I have no major objections with regards to the inttoduciton, methods, or results. I have minor comments which are mentioned below.

My primary “major” objection is the comparison of muscle forces between patient groups and conditions. The muscle activations themselves are dependant, mathematically, on the joint moments. As what you are modelling is a PTA, theoretically changes to the moment arms of these muscles are also possible and likely. However in the model you used, this could not / was not modelled. In this regard I think the focus should lay on the calculate muscle excitations/activatiosn which are you are able to compare to your EMG data, giving more confidence in your findings. You can then infer changes in muscle forces, however, a direct statement is not warranted due to the methods used. In this respect I would like to see figure 3 changed or supplemented with a comparison of muscle activations between patients, and compare to their respective EMG values.

With respect to your limitation, while I agree with what you have stated, a comment regarding not modelling the actual PTA in your model, and associated limitations in muscle forces is warranted. Further in your statement regarding patient specific- the modelling of the surgery could be added.

Purely from interest – you mentioned subjects had to have X-ray but I cannot see where you included this in the study ? Why was is required ?

Minor Spelling/Grammar

Line 19 :  but it did not change in the – rm “it did”

Line 23:  restoration of crouch gait. – shouldn’t it be reduction or treatment of crouch gait ?

Line 33: should be muscle function

Line 36: should be deficits not deformities

Line 42: When you say in respect to the severity- do you mean in proportion to severity

Line 51; Should be function not functioning

Line 63: substantially is not the right word here

Results – It would be beneficial to show the knee joint angle (flexion-extension) for both groups and time points – could be added to figure 2.

Author Response

RESPONSE TO REVIEWER 1

General comment of Reviewer 1: I would like to commend the authors on a very well designed, executed and interpreted study. I have no major objections with regards to the introduction, methods, or results. I have minor comments which are mentioned below.

Authors’ response: Thank you for the positive feedback on our study.

Reviewer comment 1: My primary “major” objection is the comparison of muscle forces between patient groups and conditions. The muscle activations themselves are dependant, mathematically, on the joint moments. As what you are modelling is a PTA, theoretically changes to the moment arms of these muscles are also possible and likely. However in the model you used, this could not / was not modelled. In this regard I think the focus should lay on the calculate muscle excitations/activatiosn which are you are able to compare to your EMG data, giving more confidence in your findings. You can then infer changes in muscle forces, however, a direct statement is not warranted due to the methods used. In this respect I would like to see figure 3 changed or supplemented with a comparison of muscle activations between patients, and compare to their respective EMG values.

Authors’ response 1: We thank the reviewer for bringing this important issue to our attention. We agree that the comparison of the model-predicted and experimental muscle activations (EMG data) would give more confidence in the interpretation of the muscle forces. In this regard, we compared the theoretical and experimental muscle activations and provided the results in the revised manuscript.

We have added the following paragraphs to the Experimental Protocol, Results, and Discussion sections. Moreover, we have added two new figures (Figure 4 and 5) in which the comparison of the experimental and model-predicted muscle activations of the PTA and NoPTA groups was provided. Furthermore, the results of the quantitative comparison (utilizing PCC values) between experimental and model-predicted muscle activations were given in a new table (Table 3).

Experimental Protocol (Page 4, lines 13-14):

…Surface EMG of the biceps femoris (BF), lateral gastrocnemius (LG), semimembranosus (SM), rectus femoris (RF), and vastus lateralis (VL) was recorded using a Myon System (Myon AG, Schwarzenberg, Switzerland). Linear envelopes of the full-wave rectified EMG signals treated with a 4th order Butterworth low-pass filter operating at a cut-off frequency of 6 Hz were calculated. X-ray measurements from lateral and medial projections were performed before and after surgery…

Results (Page 6, lines 17-22):

To evaluate the accuracy of the model-predicted muscle forces of the patient groups, we compared the model-predicted muscle activations and experimentally obtained EMG data (Figure 4 and Figure 5). The magnitudes of EMG data were normalized to each corresponding peak value. It can be revealed from Figure 4 and Figure 5 that the model-predicted muscle activation patterns were generally consistent with experimental EMG signals.

Results (Page 7, lines 1-9): (Figures 4 and 5 in extra file)

Normalized muscle activations

(a)

(b)

Figure 4. Muscle activation patterns of the PTA group. (a) preoperative and (b) postoperative muscle activations. BF: Biceps femoris, LG: Lateral gastrocnemius, RF: Rectus femoris, SM: Semimembranosus, VL: Vastus lateralis. PTA: Patellar tendon advancement, Experimental: Linear envelope of the experimental EMG signal of the PTA group normalized to each corresponding peak value, Model-predicted: Muscle activation of the PTA group predicted by static optimization.

Normalized muscle activations

(a)

(b)

Figure 5. Muscle activation patterns of the NoPTA group. (a) preoperative and (b) postoperative muscle activations. BF: Biceps femoris, LG: Lateral gastrocnemius, RF: Rectus femoris, SM: Semimembranosus, VL: Vastus lateralis. NoPTA: without patellar tendon advancement, Experimental: Linear envelope of the experimental EMG signal of the NoPTA group normalized to each corresponding peak value, Model-predicted: Muscle activation of the NoPTA group predicted by static optimization.

PCC values between the calculated muscle activations and their experimental EMG equivalents were given in Table 3. The least PCC value (0.32) was obtained from the VL muscle of the NoPTA group preoperatively, while the highest PCC value (0.95) was acquired from the RF muscle of the PTA group postoperatively.

Results (Page 8, lines 1-5):

Table 3. Mean PCC values between model-predicted muscle activations and experimental EMG data.

BF

LG

RF

SM

VL

Preoperative

PTA

0.89

0.81

0.75

0.92

0.34

NoPTA

0.76

0.84

0.85

0.77

0.32

Postoperative

PTA

0.93

0.92

0.95

0.93

0.88

NoPTA

0.77

0.94

0.91

0.73

0.86

PTA: Patellar tendon advancement, NoPTA: without patellar tendon advancement, BF: Biceps femoris, LG: Lateral gastrocnemius, RF: Rectus femoris, SM: Semimembranosus, VL: Vastus lateralis.

Discussion (Page 10, lines 35-38):

Experimental EMG activities were used to evaluate the accuracy of the model-predicted muscle activations and muscle forces (Figure 4, Figure 5, and Figure 6). It can be revealed that time-histories of the model-predicted muscle activations and forces of all five muscles agreed well with experimental EMG data.

Reviewer comment 2: With respect to your limitation, while I agree with what you have stated, a comment regarding not modelling the actual PTA in your model, and associated limitations in muscle forces is warranted. Further in your statement regarding patient specific- the modelling of the surgery could be added.

Authors’ response 2: We agree with the reviewer and added the following sentence to the Discussion section as given below.

Discussion (Page 11, lines 22-24):

…Patient-specific models including the musculoskeletal deformities would improve the accuracy of the results. Especially the modeling of PTA surgery by representing the preoperative and postoperative patella positions on the musculoskeletal model would play a critical role in the calculation of the muscle forces. Second, generic model …

Reviewer comment 3: Purely from interest – you mentioned subjects had to have X-ray but I cannot see where you included this in the study ? Why was is required?

Authors’ response 3: Thank you for the notification. The reviewer is right that we mentioned X-Ray data in the patient selection process, but not presented this data in the study. Although we did not present in the original version of the manuscript, we had calculated Insall-Salvati (IS) ratio of patients with CP using X-Ray data to determine the location of the patella pre- and postoperatively. In the revised version of the manuscript, Reviewer 3 asked us to provide IS ratio. In this regard, we provided IS ratio of the patients in the revised version of the manuscript and now addressing the use of X-Ray data in the patient selection process makes sense.

Reviewer comment 4: Line 19 :  but it did not change in the – rm “it did”.

Authors’ response 4: Done.

Abstract (Page 1, lines 18-20):

Knee extensor moment increased in the PTA group postoperatively (p=0.016), but there was no statistically significant change in the NoPTA group (p>0.05).

Reviewer comment 5: Line 23:  restoration of crouch gait. – shouldn’t it be reduction or treatment of crouch gait?

Authors’ response 5: Done.

Abstract (Page 1, line 23):

…PTA was found to be an effective surgery for the treatment of crouch gait.

Reviewer comment 6: Line 33: should be muscle function.

Authors’ response 6: Done.

Introduction (Page 1, lines 31-33):

…Crouch gait, a common pathological gait pattern in patients with CP, is characterized by abnormal kinetics, kinematics, and muscle function [3-6].

Reviewer comment 7: Line 36: should be deficits not deformities.

Authors’ response 7: Done.

Introduction (Page 1, lines 34-36):

…Patellar tendon advancement (PTA) is generally added to the surgery design of the patients with CP and crouch gait because of their knee flexion deficits [1,5,7].

Reviewer comment 8: Line 42: When you say in respect to the severity- do you mean in proportion to severity?

Authors’ response 8: Done.

Introduction (Page 1, lines 42-43):

…Brandon et al. reported that knee flexion moment significantly increased with the proportion of severity of crouch gait [12].

Reviewer comment 9: Line 51; Should be function not functioning.

Authors’ response 9: Done.

Introduction (Page 2, lines 7-9):

…Even though the function of the corresponding muscles during crouch gait is patient-specific, certain muscles are generally the target of operations for the treatment of crouch gait [14-16].

Reviewer comment 10: Line 63: substantially is not the right word here.

Authors’ response 10: Done.

Introduction (Page 2, line 19):

The effect of PTA on knee joint kinematics was shown in previous studies [1,9,11,1]. However, knowledge about …

Reviewer comment 11: Results – It would be beneficial to show the knee joint angle (flexion-extension) for both groups and time points – could be added to figure 2.

Authors’ response 11: In accordance with the comment of the reviewer, we provided knee joint angles of the PTA and NoPTA groups in the Appendix. We also revised the corresponding sentences of the Introduction section as seen below.

Introduction (Page 1, lines 39-42):

…Several studies investigated the postoperative results of the PTA using gait analysis [1,8-11]. All these studies have shown that joint kinematics, especially knee joint kinematics, improved postoperatively (please see Figure A1).

Appendix A. (Page 12, lines 10-15):

In our previous study, we investigated the effect of PTA on the knee joint kinematics of patients with CP and crouch gait [11]. We found that the postoperative knee kinematics of the patients with PTA (PTA group) was closer to that of the age-matched healthy individuals compared to the patients without PTA (NoPTA group) (Figure A1). (in extra file)

(a)

(b)

Figure A1. Mean knee joint flexion/extension angle over a gait cycle for (a) preoperative and, (b) postoperative periods. Grey zone indicates normative data obtained from the age-matched healthy reference group. PTA: Patellar tendon advancement, NoPTA: without patellar tendon advancement.

Reviewer 2 Report

There have been studies in this topic in past, but worth review and different perspectives 

would remind caution that no recommendation should be PTA alone, as the authors state these are with SEMLs on 

Overall, the paper is well written and discussed not an original topic but one that is presented nicely by the authors. It is challenging in SEMLS surgery to isolate surgical interventions and the bias of why to perform those procedures in and of itself perhaps relates to the differences in muscle tensions causing the deforming forces and deformities. Longterm follow up is always helpful for further understanding and would ask for clarity considering very involved higher level poor functioning patients with assistive devices to affect findings vs very mobile patients which may be related to the severity of the overall neuromotor dysfunction. Also, had any patients had dorsal rhizotomy history? How does that affect since the cohort if included would be heterogeneous?  In general a paper very much worth
reporting and appreciate the authors efforts and work

Author Response

RESPONSE TO REVIEWER 2

General comment of Reviewer 2: There have been studies in this topic in past, but worth review and different perspectives would remind caution that no recommendation should be PTA alone, as the authors state these are with SEMLs on. Overall, the paper is well written and discussed not an original topic but one that is presented nicely by the authors.

Authors’ response: Thank you very much for your positive feedback on our study.

Reviewer comment 1: It is challenging in SEMLS surgery to isolate surgical interventions and the bias of why to perform those procedures in and of itself perhaps relates to the differences in muscle tensions causing the deforming forces and deformities. Longterm follow up is always helpful for further understanding and would ask for clarity considering very involved higher level poor functioning patients with assistive devices to affect findings vs very mobile patients which may be related to the severity of the overall neuromotor dysfunction.

Authors’ response 1: Thank you for the comment. As the reviewer pointed out, we have discussed the limitation on the exclusion of people using assistive devices from the study. We added the following highlighted part to the Discussion section in the revised manuscript.

Discussion (Page 11, lines 30-36):

…Third, since the patients with CP show a wide heterogeneity in terms of functional capabilities, musculoskeletal deformities, and treatment histories, we applied a rigorous matching process for inclusion and exclusion of the patients, which led to a limited number of patients in the study. For example, the patients using assistive devices during pre- or postoperative exams were excluded since the use of these devices interfere with the ground reaction force, and hence, joint moments. Under these circumstances, model-predicted muscle forces may not reflect the actual situation for the patients with assistive devices. However, the effect of PTA in combination with SEMLS on the muscle forces of CP patients using assistive devices should also be investigated through subject-specific musculoskeletal models. Lastly, we provided …

Reviewer comment 2: Also, had any patients had dorsal rhizotomy history? How does that affect since the cohort if included would be heterogeneous?

Authors’ response 2: We have reviewed the operation history of the patients included in this study and found that none of these patients had dorsal rhizotomy history. Therefore, we can not comment on how dorsal rhizotomy surgery affects the postoperative outcomes of the PTA and NoPTA groups.

Reviewer comment 3: In general a paper very much worth reporting and appreciate the authors efforts and work.

Authors’ response 3: Thank you for your positive feedback.

Reviewer 3 Report

This paper looked at the effect of patellar tendon advancement in CP crouch gait patients. However, main problems are that the authors did not consider the foot and ankle status before and after surgery. This is critical as the ankle plantar flexor - knee extensor couple is the main determinant for the effectiveness of any kinds of surgery for the correction of crouch gait.

I would strongly suggest that the authors provide the following data. 

  1. The radiologic degrees of patella alta before and after surgery, such as Koshino index or any indices.
  2. Any differences in the degrees of patella alta between the cases of tendon shortening and tibial tubercle advancement?
  3. Provide additional data showing simultaneous surgeries done at another level during SEMLE?
  4. Most importantly, please provide the kinematic and kinetic data representing the ankle and foot status before and after surgery.

Author Response

RESPONSE TO REVIEWER 3

Authors’ response: Please find all figures which are ammended but which cannot be edited in this feedback form along with the full reply to all reviewers in an extra file.

Reviewer comment : This paper looked at the effect of patellar tendon advancement in CP crouch gait patients. However, main problems are that the authors did not consider the foot and ankle status before and after surgery. This is critical as the ankle plantar flexor - knee extensor couple is the main determinant for the effectiveness of any kinds of surgery for the correction of crouch gait.

I would strongly suggest that the authors provide the following data.

Authors’ response: The reviewer marks an important point, thank you. The ankle plantar flexor-knee extensor couple is a critical biomechanical issue in understanding the relationships between the knee and ankle postures. The ankle position has a substantial role in the assignment of the knee position. In the case of weak plantar flexor muscles, the ankle generates a plantar flexor moment that maintains the ground reaction force vector anterior to the knee joint. In this way, the knee joint changes its position to be more flexed and prevents itself from collapsing. Therefore, a weak plantar flexor mechanism contributes to crouch gait. (Rodda and Graham 2001; Hicks et al., 2007). Therefore, it would be valuable to evaluate the joint angles and joint moments at the ankle joint in the context of SEMLS, since the ankle dorsi/plantar flexor - knee flexor/extensor couple is the main determinant for the effectiveness of the surgeries for the correction of crouch gait. We have followed the reviewer’s recommendations and provided the required data as seen in the following comments.

Rodda, J.; Graham, H.K. Classification of gait patterns in spastic hemiplegia and spastic diplegia: a basis for a management algorithm. Eur. J. Neurol. 2001, 8, 98–108. DOI: 10.1046/j.1468-1331.2001.00042.x

Hicks, J.; Arnold, A.; Anderson, F.; Schwartz, M.; Delp, S. The effect of excessive tibial torsion on the capacity of muscles to extend the hip and knee during single-limb stance. Gait Posture 2007, 26, 546-52. DOI: 10.1016/j.gaitpost.2006.12.003

Reviewer comment 1: The radiologic degrees of patella alta before and after surgery, such as Koshino index or any indices.

Authors’ response 1: Verhulst et al. reported the Insall-Salvati (IS) ratio is the most reliable among other indices (Verhulst et al., 2020). Accordingly, we have added a new table (Table A1) to the Appendix section that included the preoperative and postoperative IS ratio of the PTA and NoPTA groups. According to the results, PTA effectively enhanced IS ratio of the patients with CP. Moreover, the bony procedure (tibial tubercle advancement) for PTA provided a better postoperative IS ratio than the soft tissue procedure (tendon shortening). We have added the following parts to the revised version of the Discussion section.

Verhulst, V.F.; van Sambeeck, J.D.P.; Olthuis, G.S.; van der Ree, J.; Koëter, S. Patellar height measurements: Insall–Salvati ratio is most reliable method. Knee Surg Sports Traumatol Arthrosc 2020, 28, 869-875. DOI: 10.1007/s00167-019-05531-1

Discussion (Page 9, lines 26-28):

In our previous study [11], we reported that PTA effectively enhanced Insall-Salvati ratio [27] (Table A1) and knee joint kinematics (Figure A1) of the patients with CP and crouch gait. The ankle plantar flexor-knee extensor couple…

Appendix A. (Page 13, lines 1-6):

In our previous study, we reported that PTA effectively enhanced Insall-Salvati ratio of patients with CP and crouch gait [11] (Table A1). Moreover, the bony procedure (tibial tubercle advancement) for PTA provided a better postoperative Insall-Salvati ratio than the soft tissue procedure (tendon shortening).

Table A1. Mean preoperative and postoperative Insall-Salvati ratios for the PTA and NoPTA groups. Significant differences were marked bold.

Preop

IS ratio

Postop

 IS ratio

PTA group

Bony (tibial tubercle advancement)

1.2 ± 0.2

0.9 ± 0.1

Soft tissue (tendon shortening)

1.1 ± 0.2

0.9 ± 0.2

Mean

1.2 ± 0.2

0.9 ± 0.1

NoPTA group

1.0 ± 0.1

0.9  ± 0.1

p-value

IS ratio

Preop PTA group vs. preop NoPTA group

0.009

Postop PTA group vs. postop NoPTA group

0.351

Preop PTA group vs. postop PTA group

0.009

Preop NoPTA group vs. postop NoPTA group

0.182

Bony procedure preop vs. soft tissue procedure preop

0.014

Bony procedure postop vs. soft tissue procedure postop

0.151

Bony procedure preop vs. bony procedure postop

0.007

Soft tissue procedure preop vs. soft tissue procedure postop

0.011

IS ratio: Insall-Salvati ratio, PTA: patellar tendon advancement, NoPTA: without patellar tendon advancement, Preop: preoperative, Postop: postoperative.

Reviewer comment 2: Any differences in the degrees of patella alta between the cases of tendon shortening and tibial tubercle advancement?

Authors’ response 2: We have provided details in Table A1. It can be seen from the table that the bony procedure (tibial tubercle advancement) for PTA provided a better postoperative Insall-Salvati ratio than the soft tissue procedure (tendon shortening).

Reviewer comment 3: Provide additional data showing simultaneous surgeries done at another level during SEMLE?

Authors’ response 3: We added a new table (Table 1) to the revised manuscript including the surgical procedures that were applied preoperatively to the patients. Please see the revised parts of the manuscript on Page 3, line 4.

Study Design (Page 2, line 50):

…Major surgeries, namely femoral extension and derotation osteotomies, hamstring lengthening, and rectus femoris transfer, applied to the patients that would affect the postoperative consequences primarily for both groups were also approved to be the same for both groups, except for the PTA surgery. The preoperative surgeries applied to the patients are provided in Table 1. In the patient selection process, …

Study Design (Page 3, lines 5-6):

Table 1. Preoperative surgeries for the PTA and NoPTA groups.

Surgical procedures

PTA group

(SEMLS with PTA)

NoPTA group

(SEMLS without PTA)

Extension osteotomy

12 patients / 21 limbs

10 patients / 14 limbs

Hamstring lengthening

7 patients / 12 limbs

6 patients / 9 limbs

Rectus femoris transfer

4 patients / 8 limbs

5 patients / 10 limbs

Derotation osteotomy

18 patients / 30 limbs

18 patients / 29 limbs

PTA: patellar tendon advancement, NoPTA: without patellar tendon advancement, SEMLS: single-event multilevel surgery.

Reviewer comment 4: Most importantly, please provide the kinematic and kinetic data representing the ankle and foot status before and after surgery.

Authors’ response 4: Thank you very much for the comment that would improve the content of this study. As the reviewer indicated we have added ankle joint moment data of the patients to the Results section. We have also provided joint angle data of the ankle of the patients in the Appendix section. Moreover, we have added the following paragraphs and sentences to the relevant parts of the Results and Discussion sections. 

Results (Page 5, lines 9-16):

The ankle joint moments of the PTA and NoPTA groups for the pre- and postoperative periods were shown in Figure 3. Magnitudes of the ankle joint moments were normalized to the mass of each patient. The preoperative patterns of the ankle joint moments of the PTA and NoPTA groups were significantly different (p=0.041) (Figure 3a). The ankle joint moment of the PTA group was closer to the healthy group than that of the NoPTA group postoperatively (p=0.046) (Figure 3b).

(a)

(b)

Figure 3. Mean plantar/dorsi flexor moment at the ankle joint over a gait cycle for (a) preoperative and (b) postoperative periods. Grey zone indicates normative data obtained from the age-matched healthy reference group. PTA: Patellar tendon advancement, NoPTA: without patellar tendon advancement.

Results (Page 6 lines  1-14):

…The RMSD and PCC values did not differ for the NoPTA group (p>0.05) postoperatively. For the ankle joint moment, the PTA group was closer to the healthy group than the NoPTA group postoperatively in terms of the RMSD and PCC values (Table 2). The mean RMSD value between the PTA and healthy groups was 0.41 preoperatively, while it was 0.66 for the NoPTA group (p=0.046). The postoperative RMSD value between the PTA and healthy groups was 0.31, and between the NoPTA and healthy groups was 0.59 (p=0.046). Between the healthy and PTA groups, the RMSD value significantly decreased postoperatively (p=0.036), but the PCC value did not differ (p>0.05). Between the healthy and NoPTA groups, the RMSD value did not change (p>0.05) postoperatively while the PPC value significantly increased (p=0.043).

Table 2. Mean RMSD and PCC values calculated between knee joint moments and ankle joint moments of CP patients and healthy subjects. Significant differences were marked bold.

Knee joint moment

Ankle joint moment

RMSD

PCC

RMSD

PCC

Preop

PTA vs. healthy

1.65

0.13

0.41

0.79

NoPTA vs. healthy

1.71

0.09

0.66

0.61

Postop

PTA vs. healthy

0.55

0.77

0.31

0.88

NoPTA vs. healthy

1.49

0.14

0.59

0.69

p-value

PTA vs. NoPTA

Preop

0.241

0.158

0.046

0.041

Postop

0.018

0.029

0.046

0.039

preop vs. postop

PTA group

0.016

0.029

0.036

0.091

NoPTA group

0.055

0.158

0.056

0.043

PTA: Patellar tendon advancement, NoPTA: without patellar tendon advancement, Preop: preoperative, Postop: postoperative. RMSD: Root-mean-square difference, PCC: Pearson cross-correlation coefficient.

Discussion (Page 9, lines 26-30; Page 10, lines 1-8):

In our previous study [11], we reported that PTA effectively enhanced Insall-Salvati ratio [27] (Table A1) and knee joint kinematics (Figure A1) of the patients with CP and crouch gait. The ankle plantar flexor-knee extensor couple is critical in understanding the relationship between the knee and ankle joints. The ankle position has a substantial role in the assignment of the knee position. In the case of weak plantar flexor muscles, the ankle generates a plantar flexor moment that maintains the ground reaction force vector anterior to the knee joint. In this way, the knee joint changes its position to be more flexed and prevents itself from collapsing. Therefore, a weak plantar flexor mechanism contributes to crouch gait [28,29]. In addition to the knee kinematics and kinetics, it would be valuable to evaluate the joint angles (Figure A2) and joint moments (Figure 3) at the anke joint in the context of SEMLS, since the ankle dorsi/plantar flexor - knee flexor/extensor couple is the main determinant for the effectiveness of the surgeries for the correction of crouch gait.

Discussion (Page 10, lines 28-34):

Ankle joint moments in the PTA and NoPTA groups significantly differed from healthy subjects preoperatively (Figure 3 and Table 2). The PTA group showed inadequate dorsiflexor moment during stance phase preoperatively, but it increased postoperatively. The NoPTA group showed excessive ankle dorsi flexor moment during the first half of the stance phase at preoperative and postoperative measurements. This finding indicated that the NoPTA group continued to have excessive dorsiflexion that may be considered as the possible cause of the crouch gait.

Appendix A. ((Page 12, lines 15-21):

Both the PTA and NoPTA groups showed excessive dorsiflexion angle at the ankle joint during stance phase preoperatively. Moreover, there was a lack of plantarflexion at the beginning of swing phase for the PTA and NoPTA groups. But for the postoperative period, the PTA group was closer to the healthy group compared to the NoPTA group. It was found that PTA surgery improved the ankle joint kinematics of patients with CP and crouch gait (Figure A2).

(a)

(b)

Figure A2. Mean ankle joint angle over a gait cycle for (a) preoperative and, (b) postoperative periods. Grey zone indicates normative data obtained from the age-matched healthy reference group. PTA: Patellar tendon advancement, NoPTA: without patellar tendon advancement.

Round 2

Reviewer 3 Report

I am happy to see the authors' reply and revised manuscript.

Author Response

Academic Editor Notes: The authors have done a good job of responding to the reviewers' comments. The paper is greatly improved after the revision. I only have two comments that require the authors' attention:

  1. Please further clarify the statistical analyses performed in the present study (Section 2.4, lines 35-45 on page 4). I was a bit confused by the structure of the statistical analyses until I saw Table 2. I would suggest creating a table to summarize the important information for the statistical analyses carried out in the present study (e.g., type of analysis, names of the dependent/independent variables).

Authors’ response: Thank you for this remark. In line 41 we added: “Statistical analysis was carried out … (RMSD and PCC values of the knee joint moment, ankle joint moment, and the muscle forces calculated between the PTA and healthy groups, and between the NoPTA and healthy groups) was evaluated by using the Shapiro-Wilk test.” and in line 48 we refer to an extra table A1 were we putline the statistical tests as was suggested.

  1. There seems to be some repeated information in the first three paragraphs of the Results section. Please consider reorganizing these three paragraphs to reduce repeatability.

Authors’ response: Thank you for this remark. In these three paragraphs we added a few more specifications regarding the comparisons which were made to clarify that there is no redundancy.